# The Root towards More Circularized Animal Production Systems: From Animal to Territorial Metabolism

**DOI:** 10.3390/ani11061540

**Published:** 2021-05-25

**Authors:** Marcello De Rosa, Jorgelina Di Pasquale, Felice Adinolfi

**Affiliations:** 1Department of Economics and Law, University of Cassino and Southern Lazio, 03043 Cassino, Italy; mderosa@unicas.it; 2Faculty of Veterinary Medicine, University of Teramo, 64100 Teramo, Italy; 3Department of Veterinary Medical Sciences, University of Bologna, Via Tolara di Sopra 50, Ozzano Emilia, 40064 Bologna, Italy; felice.adinolfi@unibo.it

**Keywords:** circular economy, sustainability, circular livestock management, agricultural policy, consumers’ acceptance

## Abstract

**Simple Summary:**

The relationship between the rates of world population growth and the consumption of natural resources is a subject of strong debate in the political and academic areas. Since the 1960s, technological progress has made it possible to achieve extraordinary increases in agricultural productivity, which was at the basis of the so-called green revolution. However, this happened at the expense of environmental sustainability. Agricultural activities impact natural resources such as water, air, biodiversity, which are crucial for future generations. The livestock sector is particularly sensitive to the problem, being responsible for an important part of the global greenhouse gas emissions. To make livestock production more sustainable, a radical rethinking of livestock production models is required. In the face of these needs, the circular economy provides a sound basis for a sustainable transition. Therefore, it is necessary to identify the crucial factors for a transition towards more “circularized” animal production systems. More precisely, our work aims to identify economic, social, and environmental factors that can boost transition, by framing it within a circular vision of livestock farming.

**Abstract:**

This paper deals with a relevant topic in the literature on sustainable management of animal farms, concerning the transition towards circular methods of animal production. The paper aims to put forward an original analytical multilevel perspective overlapping different dimensions at either micro, meso, and macro level. Starting from the Malthusian analysis on depletion of natural resources, with risks of the fragility of the natural and economic systems, the paper points out the importance of moving away from intensive methods of production, by adopting more circularized approaches based on resources efficiency. The application of circular economy approaches to animal production is theorized through the concept of territorial metabolism involving not only internal resources (at the animal farm level) but also territorial resources. The paper underlines the critical points of the transition, which is labeled as a socio-technical transition in that it involves not only technical issues but also social aspects. Critical points are addressed through consumers’ acceptance of products drawn on circular approaches and political support to transition, through political tools which are boosted in recent documents of the European Union, like the Green Deal and Farm to Fork strategy.

## 1. Introduction

Concerns about the sustainability of economic growth have fueled scientific and political debate for over two centuries. Thomas Malthus in 1798 published his work “An Essay on the Principle of Population” [1], in which he theorized the progressive fragility of the balance between the rates of population growth and the stock of available natural resources. According to Malthus, without a policy of birth control, the balance between consumption and regeneration of agricultural resources would be undermined, and in order to restore it, temporary disruptions of economic development due to famine or epidemics would happen [1]. This view of the relationship between the availability of agricultural resources and population growth has lost its importance in the following decades. However, it has only recently come back into vogue after the food price spikes of 2007 and the alarms launched by the World Bank [2] and other international agencies about the challenge of meeting the rising demand for food over the next thirty years [3,4,5].

Technological progress has made it possible to achieve extraordinary increases in agricultural productivity by removing at least part of the problem of the availability of fertile land. In particular, since the sixties, the articulated set of new input-sensitive seeds, irrigation techniques, fertilizers, and pesticides [6], which was at the basis of the so-called green revolution, has triggered a progressive growth of agricultural yields. However, this happened at the expense of environmental sustainability and the neo-Malthusian nightmare [7] today has a broader focus, involving a wide range of natural resources. Agricultural activities affect many fundamental resources such as water, air, biodiversity, all of which are crucial for future generations [8].

The livestock sector is particularly sensitive to the problem, being responsible for a part of global greenhouse gas emissions. Moreover, its intensity of consumption of natural resources—water and agricultural products for animal feed—is higher than other agricultural production. The challenge of matching the demand for animal products resulting from both the rising population and the convergence of dietary styles [9] requires a radical rethinking of livestock production models [10]. The circular economy provides a sound basis for a sustainable transition.

Against this background, this paper questions key issues concerning the potential for a transition towards more “circularized” animal production systems. More precisely, our work aims to identify the factors that are crucial for an either economically, socially, and environmentally sound transition, by framing it within a circular vision of livestock farming.

The paper is articulated as follows: Section 2 of the paper discusses issues of the transition from a linear to a circular economy on animal farms as they move from conventional models of animal production (i.e., intensive livestock production) to more sustainable livestock production. Section 3 discusses systemic perspective involving consumer behavior as a promoter and supporter of circular and sustainable production systems. Section 4 calls for policy action in the event of market failure related to lack of support and recognition by consumers. Preliminary conclusions are the final portion of the paper.

## 2. From Linear to Circular Approaches in Small/Medium Scale Animal Farms

In this section, we put forward an approach able to replicate at the territorial scale the sustainable management approaches at the basis of the animal metabolism. More precisely, we build up a territorial approach, through embedding animal production in territorial contexts through a multilevel perspective.

Recent literature has widely underlined the limits of conventional models of animal production [11,12,13]. As a matter of fact, breeding and livestock activities are essentially grounded on linear mechanisms of production [14,15]. However, in the linear perspective, negative externalities emerge, whose cost is sustained by actors other than producers. This is particularly true in intensive livestock and slaughtering sectors: in fact, the linear economy model based on take-make-use-dispose has brought about negative consequences on the animal sector, in the account of a large amount of input used to feed the intensive model of animal production. The negative effects, that is, negative externalities produced by intensive livestock have engendered higher social than private costs, with negative consequences on the entire society.

On the other side, a circular economy approach is “a systems-level approach to economic development designed to benefit businesses, society, and the environment” [16]. More precisely, a circular economy searches for a balance between economic growth and environmental issues, through efficient resource utilization and recycling. 

The role of the animal sector in boosting circular food systems is fundamental and could be both indirect and direct. Indirect effects include a sensible reduction in the consumption of animal-source food that has positive impacts on the environment [17,18,19]. Direct effects include the transition to a circular economy which may happen in the animal sector through a minimization in the use of external inputs and by reducing wastes and emissions in the environment thanks to recycling and valorization of agricultural wastes, as pointed out in an EU position paper [20,21,22].

Recent research has evidenced the great opportunities the circular economy offers to animal farms, by using “animals for what they are good at” [10] (p. 20). As a consequence, this encourages processes of paradigm shift, in that the transition is realized through the use of local resources, by preserving biodiversity and putting forwards sustainable models of livestock [23]. In this paper we posit that similar to animal metabolism, grounded on energetic inputs allowing the animal to sustain itself, sustainable management of territorial resources may bring about a sustainable process of ecological transition, which involves social, economic, and environmental aspects, bringing about what we label here as territorial metabolism.

Accordingly, to properly address circular economy issues in animal farming, it is of paramount importance to clarify the three interrelated dimensions of circularity: social, technical/economic, and environmental. First of all, from a social point of view, regenerative farming is grounded on resiliency and on multifunctionality [24], which implies more a socio-cultural transition than a mere technological step. This transition breaks the mold with the past ways of doing things. From an economic point of view, the alternative provided by the circular economy may bring about positive economic results. It is necessary to perceive a circular economy not only as a necessity but as an opportunity, that may provide the farming sector with quantifiable economic advantages. Recent data presented by the most important Italian Trade Union demonstrate how circular economy has generated a turnover of USD 88 bln in 2019 in Italy [25]. Finally, from a technical point of view, a circular economy implies a transition towards new technologies. Sustainable intensification is the new paradigm involving a trend of maximization of yields, by maintaining the sustainability of agricultural processes. A typical example is offered by smart farming and precision agriculture, which are gaining importance as technologies allowing saving natural resources. Precision farming is defined as “a farming management concept based upon observing, measuring and responding to inter and intra-field variability in crops or in aspects of animal rearing” [26] (p. 11). It is increasingly applied not only to crop systems but to the whole management of farms (animal farms included). More precisely, precision farming has revealed effectiveness in animal farming through minimization in the use of fertilizer and agrochemical inputs, based on soil variability. Furthermore, precision livestock farming is of fundamental importance for monitoring animal behavior, welfare, and productivity, and also animals’ physical environment [26,27]. As pointed out by Ward et al. [21] (p. 4), “while not directly contributing to ‘circularisation’, precision livestock farming addresses the use of minimal levels of invested resources that is essential to achieving sustainable agricultural production”.

Another interesting example is aquaponics, which is an effective model of circular economy, thanks to the capability of recreating natural ecosystems in artificial environments [28] and challenging land, nutrient, and water scarcity, and reducing energy use and food miles [29].

Moreover, innovation paths able to support the role of animal farms in food systems act along with processes of paradigm shift, which has been labeled as socio-technical transition, in that it also involves social aspects [30]. Indeed, societal acceptance becomes the engine of innovation processes in animal farming towards agroecological transition, where key variables become added value/unit of labor, instead of maximizing output/input [31].

Nonetheless, as with every novelty, the transition involves a “deviation from the rules”, it requires time to break with existing rules and, in many cases, may also fail [32]. As a matter of fact, a rigorous analysis of the transition process calls for multilevel perspectives, to take into account all variables involved in the process.

### A Multilevel Perspective on Circularization in Animal Production

To boost a successful transition, a circular economy may offer a sound answer, whose basic principle is self-sufficiency to be realized through interrelating multiple dimensions, under a multilevel (animal, farm, supply chain/market, policy/decision-makers) and multiscale (local, regional, national, global) perspective. Efficiency is measured through energetic autonomy within each functional and territorial space; whereas, energetic autonomy is provided by the net balance in the interaction between production and consumption of natural resources.

Focusing on animal production systems, a multilevel perspective allows to the evaluation of the following overlapping analytical levels:(a)the first functional micro-level is the animal, more precisely its metabolic efficiency, which affects energy exchange between animal and natural resources [33,34];(b)the second analytical node concerns the animal farm, whose organization of production inputs may affect the outcome of biophysical exchanges which are drawn on this functional territorial space [35,36];(c)moving towards a more complex analytical level, the market has to be considered as the functional space allowing the ascertainment of the contribution of every single product or service to circularity building.

Crossing functional dimensions over territorial scales may provide sounder analyses, with special reference to animal systems of production: the strong connections with, on the one side, land structures and, on the other side, with high specialization of productive systems [37,38] have opened new space for territorial analysis of the food supply chains, as demonstrated in numerous studies investigating biophysical flows in the agri-food systems [39], concerning the concept of “socioeconomic metabolism” [40,41]. This paradigm is drawn on Georgescu-Roegen’s [42] model aimed to represent metabolic processes mediated by men through concepts like stock, funds, and flows. Moreover, it helps to research how energy exchanges affect the evolution of biophysical structures of society [40,43].

The application of this concept on a territorial scale has been analyzed in the literature on sustainability of agrifood systems [44,45,46] and transformation of rural space [47,48], by showing its ability to catch the role of systemic components in metabolic performance. If we assume territory as intermediate space (meso level) between the micro (animal and farm) and macro (market) components in the analysis of the metabolism of animal production systems and if we assume circularity to evaluate their energetic self-efficiency, the biophysical flows system may be represented as a system of “metabolic nodes”, whose energetic autonomy affects the upward larger node, the final result has an impact on the stock of available resources (Figure 1).

Nonetheless, it cannot be neglected that more circular systems of animal production engender costs of transition that must be “compensated”. Therefore, two main aspects deserve attention to support transition:(a)consumers’ recognition and appreciation for the benefits of circular economy and sustainable management practices, bringing about a premium price and higher availability to consume products from circular economy (market mechanisms);(b)in case of non-recognition by the consumer, that is market failure, policy support towards farms willing to adopt this socio-technical transition (policy mechanism).

These two aspects will be analyzed in the following sections.

## 3. Changes in Consumers’ Preferences as a Driver of Sustainable Management Practices

Economic performance in the choice of sustainable management practices, like circular approaches, is boosted by changes in consumers’ preferences and behavior. A recent literature review by Camacho-Otero et al. [49] highlights different factors driving the consumption of circular solutions. Consumers’ acceptance seems to depend on personal characteristics (i.e., personality traits, values, and ideologies that may influence consumer perceptions), level of knowledge about and understanding of a specific product, consumption experience, and its impact on everyday life. Moreover, in a recent report, Kirchherr et al. [50] found that the lack of consumer interest and awareness is a “main impediment regarding a transition towards circular economy”. Likewise, Rizos et al. [51] (p7) posit that the “lack of support from demand networks” prevented the implementation of green innovations such as circular business models. In this perspective, the consumer is both a promoter and an obstacle to the transition, then questioning “compatibility” issues with the new patterns of consumption [52].

Radical marketing is at stake here, where producers put forwards new ethical values of both production and consumption [53,54]. As pointed out by Brunori [55] (p.3), the common feature of these initiatives is the involvement of values of social, ethical, and environmental aspects that are recognizable by consumers and put into business. This lets co-production mechanisms between consumer and producer emerge, bringing about more “circularized” systems of agricultural production [56,57].

Populations in advanced economies that have already reached the point of satiety are turning their attention to attributes that go beyond nutritional levels and food safety, leading the food sector towards ethical choices within a circular economy vision.

Globally, in the last decades, consumers have paid particular attention to products deriving from sustainable and organic agriculture [58], which are based on two fundamental ethical principles, environmental protection and animal welfare.

Drivers of food innovations in Europe can be divided into 15 trends, grouped along five axes, corresponding to general consumer expectations: pleasure, health, physical, convenience, and ethics. The health component is one of the first three main components that guide consumers’ choices. Although ethical attributes are the fifth of the five drivers (in percentage), it remains the most dynamic driver in food innovation in Europe in terms of growth [59].

Not just in Europe, but all over the world, more attention is paid to the ethical aspects concerning both social and ecological issues. Growing consideration is given to the protection of the territory and the respect of animal welfare [60]. This brings about abandoning a productivist perspective of the animal to reach a more complex vision of balance between nutritional needs and respect for the territory and the environment. However, these requests are not always accompanied by a consumers’ willingness to pay for compensating the efforts (and the higher costs) of the production sector. Furthermore, the consumer is not always able to recognize the products obtained through compliance with higher ethical standards. These factors can lead to market failure, in these situations policies need to be ready to take action.

The rise of per capita income, urbanization, but also female labor participation, economic globalization, and social and meat prices, are at the root of the motivations that are driving global changes and the demand for protein foods of animal origin worldwide [61]. These dynamics will continue over the years and this implies the need for global attention to avoid an intensification of production at the expense of sustainability and respect for animals. Respect that should pass not only from ensuring the best possible level of welfare for animals but also from avoiding food losses and waste. The abandonment of linear production in favor of circular production on a global level and not only in advanced economies.

In light of these new needs, the "sustainable intensification" concept was defined [62]. The global food trends indicate that animal production must be supported, however, it must be carried out in a sustainable way and also taking into account the welfare of animals. Sustainable intensification theorists argue that when animal welfare is compromised, there are significant negative consequences on human health, due to environmental degradation, the use of non-therapeutic levels of antibiotics for growth promotion, and the consequences of intensification [62].

Therefore, improving a circular approach that includes animal welfare is no longer just an ethical prerogative, but a health requirement with direct and indirect repercussions on human life.

As recognized in the “One Health Approach”, the health of animals, people, plants, and the environment is interrelated. As a consequence, health problems must be dealt with together [63]. To achieve better public health outcomes, a multidisciplinary approach that mutually integrates policies, legislation, and research in communicating and working together is needed.

Future food policies must think simultaneously and in an overlapping manner for both the agricultural and health sectors, so as to be able to develop coherent and sustainable policies that respond to the needs of populations both from the point of view of food safety and health, human and environmental health.

## 4. The Political Dimension in Addressing Sustainable Management in Animal Farms

The transition towards a circular economy may be hindered by market failures, which call for policy action. Resource scarcity, population growth, and new environmental challenges are leading politicians to rethink development trajectories.

Policy action is targeted towards the proposal of new circular business models, aiming at improving quality of life through new value propositions and sustainable management of animal farms [64].

Circular business models in animal farms are encouraged by the *European Green Deal* and by the *Farm to Fork* strategy, whose main objectives are: to enhance the use of renewable energy;Reduction in the use of pesticide before 2030;Incentives for sustainable and organic farming;Support of animal welfare.

The final purpose is to support a sustainable economy in the EU, through “turning climate and environmental challenges into opportunities”. As posited by the European Commission, *it is clear that the transition must be supported by a CAP that focuses on the Green Deal* [65] (p. 9).

In the last decades, common agricultural policy (CAP) has integrated various dimensions in the sphere of increasing productivity and encouraging competitiveness through preserving the sustainability of animal systems. Accordingly, policy intervention has privileged “contractual” approaches, by subordinating funding to institutional arrangements preserving animal welfare, sustainable animal production, adoption of greening practices, etc.

European agriculture is on the right root, as it is possible to date back the policy for circular economy in animal production since 1996, with the document “Agenda 2000”, which has officially launched the new European agricultural model, based on multifunctional agriculture, low farming intensity and productivity [66]. The purpose was to promote a new European agricultural model which positively impacts the environment, benefitting society and providing high-quality goods [67]. The following revisions of the CAP have acknowledged the call for reducing the impact of animal production on climate change, by fostering more extensive practices and reducing waste production. As a matter of fact, *agriculture, forestry, and land use account for at least 20% of total emissions, mainly from the conversion of forests to farmland and from livestock and crop production* [68]. Moreover, animal production is considered to have an impact on climate change: among the largest emitters in agriculture are enteric fermentation (40%) and manure left on pastures (16%) [68]. On the other side, animal production is affected by climate change in terms of higher vulnerability of animal production systems, due to increased frequency of dry spells and drought, changes in precipitation patterns, increasing intensity of extreme weather events, and rising temperature [69].

Responding to climate change brings about adopting sustainable agricultural practices. This is particularly true in animal farming, where the effects of climate change may be relevant in terms of water availability, impacts on pasture and forage crop quantity and quality, etc. [70]. Policies for agricultural and livestock innovation have distanced themselves from productivist approaches based on agricultural intensification, by orienting innovation towards bottom-up approaches where productivity is strictly joined with sustainability. For instance, in the case of the European innovation partnership for agricultural innovation (EIP-AGRI), a relevant part of funded innovations involves efficient use of resources and climate change and preservation/valorization of agricultural ecosystems.

## 5. Conclusions

In this paper, we have tried to shed light on a critical point for supporting sustainable management in animal production. More precisely, starting from the environmental challenges emerging in the last decades, due to intensive methods of production, we have put forward an original approach drawn on a multilevel perspective, which overlaps micro/meso/macro dimensions (concerning the animal, the farm, and the market) with the institutional scale of analysis (involving international, national and regional scales). This has brought about the adoption of a territorial metabolism perspective, which represents a condition sine qua non for boosting (sociotechnical) transition towards a circular economy in the animal sector. Each level of analysis offers insight for assuming a perspective of circular economy, to improve the energetic efficiency and, consequently, reducing the impact on climate change, improving resource management, and, finally, the efficiency of animal farm management.

The connections between the subsystems involved become more and more complex as we move from the micro to the macro level of this framework and conflict resolution is strongly influenced by policies. The ambitions of the Green Deal are not pursued with the setting of environmental standards but through a system of political measures that work in an integrated way to support the ecological transition, which also implies a review of the economic and social vision of the future. To be realized, the contribution of international policies and in particular of more efficient regulatory mechanisms in the governance of the macro level (the market) is required, jointly with the recognition of the ecological footprint by the consumer.

## Figures and Tables

**Figure 1 animals-11-01540-f001:**
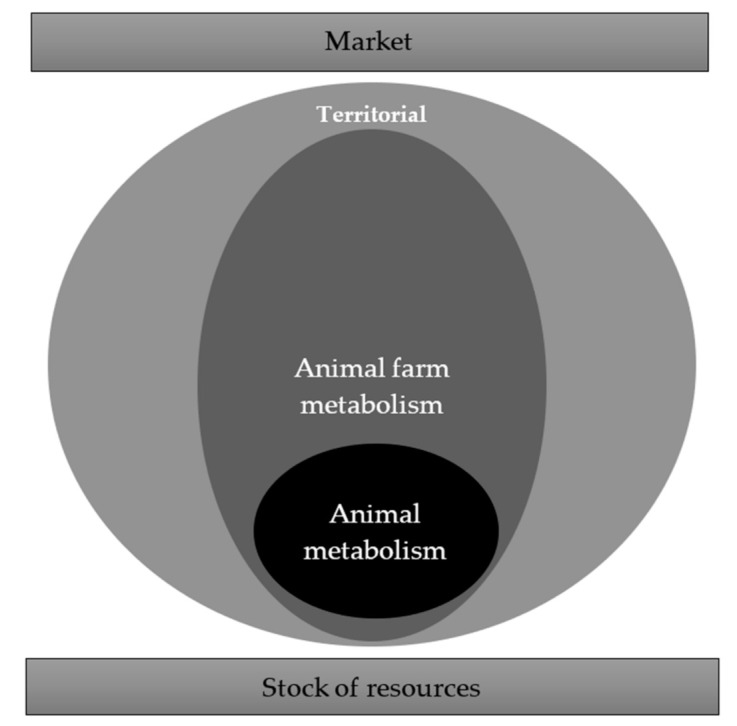
Functional spaces and circular economy in animal production.

## Data Availability

Not applicable.

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
