# Peer review of "The Root towards More Circularized Animal Production Systems: From Animal to Territorial Metabolism"

_animals, 2021, doi:10.3390/ani11061540_

Round 1

Reviewer 1 Report

Potentials to improve the sustainability of livestock farming are a current and controversially discussed topic. The paper takes up this relevant topic and discusses it from a multilevel perspective. I very much welcome the fact that the authors explicitly consider animal welfare as an important component for sustainable livestock production and also consider the importance of consumers. From my perspective, however, it would be helpful if the authors could provide one or two more examples of a circular approach in addition to the precision farming example. Perhaps aquaponics could be mentioned here, which might come close to the concept described by the authors if consumers are involved (e.g. in the context of urban/social farming concepts). 

Specific notes:

I would reconsider the word "circularized". At least my dictionaries and translators translate this exclusively in the sense of "made known" and not in the sense of circular.

Lines 24/25 and 31/32 have the same content

Line 74: Instead of animal protein, I would use the term animal products because animal products do not only contain proteins as important components, but also some important micronutrients.

Line 261: The dot after "origin" should be deleted

Line 336: "methods" (instead of "method")

Author Response

Thank you very much, we agree with you and have provided the suggested example which allowed to improve the quality of the paper. 

Reviewer 2 Report

I read with interest the concept paper "The root towards more circularized animal production systems:  from animal to territorial metabolism" and I think that the presented indication is very interesting and useful to understand synthetically the transition period in the animal production system. The paper is well written, clear and well explained.
The introduction is clear and provides information due to understand the objectives of the study.
“Consumers’ preferences” and “political dimension” paragraph are well-explained but in my humble opinion need more reference to boost the proposed concepts. Therefore, I suggest to improve the literature review with more recent bibliography should be presented. 
The Conclusion is in line with the contents of previous sections and I agree with it.  
In a whole, I appreciate the topic of the paper and I consider the study interesting and quite original. Therefore, in my humble opinion, it could be eligible for the publication on IF journal as Animals, only after minor revision.

Author Response

Dear reviewer, thanks a lot for your comments, we have modified the paper according to your suggestion. We think the revisions allowed to improve the quality of the paper.
